# A Non-generative Framework and Convex Relaxations for Unsupervised Learning

**Elad Hazan**
Princeton University
35 Olden Street 08540
ehazan@cs.princeton.edu.

**Tengyu Ma**
Princeton University
35 Olden Street, NJ 08540
tengyu@cs.princeton.edu.

## Abstract

We give a novel formal theoretical framework for unsupervised learning with two distinctive characteristics. First, it does not assume any generative model and based on a worst-case performance metric. Second, it is comparative, namely performance is measured with respect to a given hypothesis class. This allows to avoid known computational hardness results and improper algorithms based on convex relaxations. We show how several families of unsupervised learning models, which were previously only analyzed under probabilistic assumptions and are otherwise provably intractable, can be efficiently learned in our framework by convex optimization.

## 1 Introduction

Unsupervised learning is the task of learning structure from unlabelled examples. Informally, the main goal of unsupervised learning is to extract structure from the data in a way that will enable efficient learning from future labelled examples for potentially numerous independent tasks.

It is useful to recall the Probably Approximately Correct (PAC) learning theory for supervised learning [28], based on Vapnik's statistical learning theory [29]. In PAC learning, the learning can access labelled examples from an unknown distribution. On the basis of these examples, the learner constructs a hypothesis that generalizes to unseen data. A concept is said to be learnable with respect to a hypothesis class if there exists an (efficient) algorithm that outputs a generalizing hypothesis with high probability after observing polynomially many examples in terms of the input representation.

The great achievements of PAC learning that made it successful are its generality and algorithmic applicability: PAC learning does not restrict the input domain in any way, and thus allows very general learning, without generative or distributional assumptions on the world. Another important feature is the restriction to specific hypothesis classes, without which there are simple impossibility results such as the "no free lunch" theorem. This allows *comparative* and *improper* learning of computationally-hard concepts.

The latter is a very important point which is often understated. Consider the example of sparse regression, which is a canonical problem in high dimensional statistics. Fitting the best sparse vector to linear prediction is an NP-hard problem [20]. However, this does not prohibit improper learning, since we can use a $\ell_1$ convex relaxation for the sparse vectors (famously known as LASSO [26]).

Unsupervised learning, on the other hand, while extremely applicative and well-studied, has not seen such an inclusive theory. The most common approaches, such as restricted Boltzmann machines, topic models, dictionary learning, principal component analysis and metric clustering, are based almost entirely on generative assumptions about the world. This is a strong restriction which makes it very hard to analyze such approaches in scenarios for which the assumptions do not hold. A more discriminative approach is based on compression, such as the Minimum Description Length

criterion. This approach gives rise to provably intractable problems and doesn't allow improper learning.

**Main results.** We start by proposing a rigorous framework for unsupervised learning which allows data-dependent, comparative learning without generative assumptions about the world. It is general enough to encompass previous methods such as PCA, dictionary learning and topic models. Our main contribution are optimization-based relaxations and efficient algorithms that are shown to improperly probably learn previous models, specifically:

1. We consider the classes of hypothesis known as dictionary learning. We give a more general hypothesis class which encompasses and generalizes it according to our definitions. We proceed to give novel polynomial-time algorithms for learning the broader class. These algorithms are based on new techniques in sum-of-squares convex relaxations.

   As far as we know, this is the first result for efficient improper learning of dictionaries without generative assumptions. Moreover, our result handles polynomially over-complete dictionaries, while previous works [4, 8] apply to at most constant factor over-completeness.

2. We give efficient algorithms for learning a new hypothesis class which we call spectral autoencoders. We show that this class generalizes, according to our definitions, the class of PCA (principal component analysis) and its kernel extensions.

**Structure of this paper.** In the following chapter we a non-generative, distribution-dependent definition for unsupervised learning which mirrors that of PAC learning for supervised learning. We then proceed to an illustrative example and show how Principal Component Analysis can be formally learned in this setting. The same section also gives a much more general class of hypothesis for unsupervised learning which we call polynomial spectral decoding, and show how they can be efficient learned in our framework using convex optimization. Finally, we get to our main contribution: a convex optimization based methodology for improper learning a wide class of hypothesis, including dictionary learning.

## 1.1 Previous work

The vast majority of work on unsupervised learning, both theoretical as well as applicative, focuses on generative models. These include topic models [11], dictionary learning [13], Deep Boltzmann Machines and deep belief networks [24] and many more. Many times these models entail non-convex optimization problems that are provably NP-hard to solve in the worst-case.

A recent line of work in theoretical machine learning attempts to give efficient algorithms for these models with provable guarantees. Such algorithms were given for topic models [5], dictionary learning [6, 4], mixtures of gaussians and hidden Markov models [15, 3] and more. However, these works retain, and at times even enhance, the probabilistic generative assumptions of the underlying model. Perhaps the most widely used unsupervised learning methods are clustering algorithms such as k-means, k-medians and principal component analysis (PCA), though these lack generalization guarantees. An axiomatic approach to clustering was initiated by Kleinberg [17] and pursued further in [9]. A discriminative generalization-based approach for clustering was undertaken in [7] within the model of similarity-based clustering.

Another approach from the information theory literature studies with online lossless compression. The relationship between compression and machine learning goes back to the Minimum Description Length criterion [23]. More recent work in information theory gives online algorithms that attain optimal compression, mostly for finite alphabets [1, 21]. For infinite alphabets, which are the main object of study for unsupervised learning of signals such as images, there are known impossibility results [16]. This connection to compression was recently further advanced, mostly in the context of textual data [22].

In terms of lossy compression, Rate Distortion Theory (RDT) [10, 12] is intimately related to our definitions, as a framework for finding lossy compression with minimal distortion (which would correspond to reconstruction error in our terminology). Our learnability definition can be seen of an extension of RDT to allow improper learning and generalization error bounds. Another learning framework derived from lossy compression is the information bottleneck criterion [27], and its

learning theoretic extensions [25]. The latter framework assumes an additional feedback signal, and thus is not purely unsupervised.

The downside of the information-theoretic approaches is that worst-case competitive compression is provably computationally hard under cryptographic assumptions. In contrast, our compression-based approach is based on learning a restriction to a specific hypothesis class, much like PAC-learning. This circumvents the impossibility results and allows for improper learning.

## 2   A formal framework for unsupervised learning

The basis constructs in an unsupervised learning setting are:

1. Instance domain $\mathcal{X}$, such as images, text documents, etc. Target space, or range, $\mathcal{Y}$. We usually think of $\mathcal{X} = \mathbb{R}^d, \mathcal{Y} = \mathbb{R}^k$ with $d \gg k$. (Alternatively, $\mathcal{Y}$ can be all sparse vectors in a larger space. )

2. An unknown, arbitrary distribution $\mathcal{D}$ on domain $\mathcal{X}$.

3. A hypothesis class of decoding and encoding pairs,

$$\mathcal{H} \subseteq \{(h, g) \in \{\mathcal{X} \mapsto \mathcal{Y}\} \times \{\mathcal{Y} \mapsto \mathcal{X}\}\},$$

where $h$ is the encoding hypothesis and $g$ is the decoding hypothesis.

4. A loss function $\ell : \mathcal{H} \times \mathcal{X} \mapsto \mathbb{R}_{\geqslant 0}$ that measures the reconstruction error,

$$\ell((g, h), x).$$

For example, a natural choice is the $\ell_2$-loss $\ell((g, h), x) = \|g(h(x)) - x\|_2^2$. The rationale here is to learn structure without significantly compromising supervised learning for *arbitrary* future tasks. Near-perfect reconstruction is sufficient as formally proved in Appendix 6.1. Without generative assumptions, it can be seen that near-perfect reconstruction is also necessary.

For convenience of notation, we use $f$ as a shorthand for $(h, g) \in \mathcal{H}$, a member of the hypothesis class $\mathcal{H}$. Denote the generalization ability of an unsupervised learning algorithm with respect to a distribution $\mathcal{D}$ as

$$\operatorname*{loss}_{\mathcal{D}}(f) = \mathop{\mathbb{E}}_{x \sim \mathcal{D}}[\ell(f, x)].$$

We can now define the main object of study: unsupervised learning with respect to a given hypothesis class. The definition is parameterized by real numbers: the first is the encoding length (measured in bits) of the hypothesis class. The second is the bias, or additional error compared to the best hypothesis. Both parameters are necessary to allow improper learning.

*Definition* 2.1. We say that instance $\mathcal{D}, \mathcal{X}$ is $(k, \gamma)$-$C$-learnable with respect to hypothesis class $\mathcal{H}$ if exists an algorithm that for every $\delta, \varepsilon > 0$, after seeing $m(\varepsilon, \delta) = \operatorname{poly}(1/\varepsilon, \log(1/\delta), d)$ examples, returns an encoding and decoding pair $(h, g)$ (not necessarily from $\mathcal{H}$) such that:

1. with probability at least $1 - \delta$, $\operatorname{loss}_{\mathcal{D}}((h, g)) \leqslant \min_{(h,g) \in \mathcal{H}} \operatorname{loss}_{\mathcal{D}}((h, g)) + \varepsilon + \gamma$.

2. $h(x)$ has an explicit representation with length at most $k$ bits.

For convenience we typically encode into real numbers instead of bits. Real encoding can often (though not in the worst case) be trivially transformed to be binary with a loss of logarithmic factor.

Following PAC learning theory, we can use uniform convergence to bound the generalization error of the empirical risk minimizer (ERM). Define the empirical loss for a given sample $S \sim \mathcal{D}^m$ as

$$\operatorname*{loss}_{S}(f) = \frac{1}{m} \cdot \sum_{x \in S} \ell(f, x)$$

Define the ERM hypothesis for a given sample $S \sim \mathcal{D}^m$ as $\hat{f}_{ERM} = \arg\min_{\hat{f} \in \mathcal{H}} \operatorname{loss}_S(\hat{f})$.

For a hypothesis class $\mathcal{H}$, a loss function $\ell$ and a set of $m$ samples $S \sim \mathcal{D}^m$, define the empirical Rademacher complexity of $\mathcal{H}$ with respect to $\ell$ and $S$ as, [1]

$$\mathcal{R}_{S,\ell}(\mathcal{H}) = \mathop{\mathbb{E}}_{\sigma \sim \{\pm 1\}^m} \left[ \sup_{f \in \mathcal{H}} \frac{1}{m} \sum_{x \in S} \sigma_i \ell(f, x) \right]$$

Let the Rademacher complexity of $\mathcal{H}$ with respect to distribution $\mathcal{D}$ and loss $\ell$ as $\mathcal{R}_m(\mathcal{H}) = \mathbb{E}_{S \sim \mathcal{D}^m}[\mathcal{R}_{S,\ell}(\mathcal{H})]$. When it's clear from the context, we will omit the subscript $\ell$.

We can now state and apply standard generalization error results. The proof of following theorem is almost identical to [19, Theorem 3.1]. For completeness we provide a proof in Appendix 6.

*Theorem* 2.1. For any $\delta > 0$, with probability $1 - \delta$, the generalization error of the ERM hypothesis is bounded by:

$$\mathrm{loss}_{\mathcal{D}}(\hat{f}_{ERM}) \leqslant \min_{f \in \mathcal{H}} \mathrm{loss}_{\mathcal{D}}(f) + 6\mathcal{R}_m(\mathcal{H}) + \sqrt{\frac{4 \log \frac{1}{\delta}}{2m}}$$

An immediate corollary of the theorem is that as long as the Rademacher complexity of a hypothesis class approaches zero as the number of examples goes to infinity, it can be $C$ learned by an *inefficient* algorithm that optimizes over the hypothesis class by enumeration and outputs an best hypothesis with encoding length $k$ and bias $\gamma = 0$. Not surprisingly such optimization is often intractable and hences the main challenge is to design efficient algorithms. As we will see in later sections, we often need to trade the encoding length and bias slightly for computational efficiency.

**Notations:** For every vector $z \in \mathbb{R}^{d_1} \otimes \mathbb{R}^{d_2}$, we can view it as a matrix of dimension $d_1 \times d_2$, which is denoted as $\mathcal{M}(z)$. Therefore in this notation, $\mathcal{M}(u \otimes v) = uv^\top$. Let $v_{\max}(\cdot) : (\mathbb{R}^d)^{\otimes 2} \to \mathbb{R}^d$ be the function that compute the top right-singular vector of some vector in $(\mathbb{R}^d)^{\otimes 2}$ viewed as a matrix. That is, for $z \in (\mathbb{R}^d)^{\otimes 2}$, then $v_{\max}(z)$ denotes the top right-singular vector of $\mathcal{M}(z)$. We also overload the notation $v_{\max}$ for generalized eigenvectors of higher order tensors. For $T \in (\mathbb{R}^d)^{\otimes \ell}$, let $v_{\max}(T) = \mathrm{argmax}_{\|x\| \leqslant 1} T(x, x, \ldots, x)$ where $T(\cdot)$ denotes the multi-linear form defined by tensor $T$.

# 3 Spectral autoencoders: unsupervised learning of algebraic manifolds

## 3.1 Algebraic manifolds

The goal of the spectral autoencoder hypothesis class we define henceforth is to learn the representation of data that lies on a low-dimensional algebraic variety/manifolds. The linear variety, or linear manifold, defined by the roots of linear equations, is simply a linear subspace. If the data resides in a linear subspace, or close enough to it, then PCA is effective at learning its succinct representation.

One extension of the linear manifolds is the set of roots of low-degree polynomial equations. Formally, let $k, s$ be integers and let $c_1, \ldots, c_{d^s - k} \in \mathbb{R}^{d^s}$ be a set of vectors in $d^s$ dimension, and consider the algebraic variety

$$\mathcal{M} = \left\{ x \in \mathbb{R}^d : \forall i \in [d^s - k], \langle c_i, x^{\otimes s} \rangle = 0 \right\} .$$

Observe that here each constraint $\langle c_i, x^{\otimes s} \rangle$ is a degree-$s$ polynomial over variables $x$, and when $s = 1$ the variety $\mathcal{M}$ becomes a liner subspace. Let $a_1, \ldots, a_k \in \mathbb{R}^{d^s}$ be a basis of the subspaces orthogonal to all of $c_1, \ldots, c_{d^s - k}$, and let $A \in \mathbb{R}^{k \times d^s}$ contains $a_i$ as rows. Then we have that given $x \in \mathcal{M}$, the encoding

$$y = Ax^{\otimes s}$$

pins down all the unknown information regarding $x$. In fact, for any $x \in \mathcal{M}$, we have $A^\top A x^{\otimes s} = x^{\otimes s}$ and therefore $x$ is decodable from $y$. The argument can also be extended to the situation when the data point is close to $\mathcal{M}$ (according to a metric, as we discuss later). The goal of the rest of the subsections is to learn the encoding matrix $A$ given data points residing close to $\mathcal{M}$.

**Warm up: PCA and kernel PCA.** In this section we illustrate our framework for agnostic unsupervised learning by showing how PCA and kernel PCA can be efficiently learned within our model. The results of this sub-section are not new, and given only for illustrative purposes. The class of hypothesis corresponding to PCA operates on domain $\mathcal{X} = \mathbb{R}^d$ and range $\mathcal{Y} = \mathbb{R}^k$ for some $k < d$ via linear operators. In kernel PCA, the encoding linear operator applies to the $s$-th tensor power $x^{\otimes s}$ of the data. That is, the encoding and decoding are parameterized by a linear operator $A \in \mathbb{R}^{k \times d^s}$,

$$\mathcal{H}_{k,s}^{\mathrm{pca}} = \left\{ (h_A, g_A) : h_A(x) = Ax^{\otimes s}, \ , g_A(y) = A^\dagger y \right\} ,$$

where $A^\dagger$ denotes the pseudo-inverse of $A$. The natural loss function here is the Euclidean norm, $\ell((g, h), x) = \|x^{\otimes s} - g(h(x))\|^2 = \|(I - A^\dagger A)x^{\otimes s}\|^2$.

*Theorem* 3.1. For a fixed constant $s \geqslant 1$, the class $\mathcal{H}_{k,s}^{\mathrm{pca}}$ is efficiently $C$-learnable with encoding length $k$ and bias $\gamma = 0$.

The proof of the Theorem follows from two simple components: a) finding the ERM among $\mathcal{H}_{k,s}^{\mathrm{pca}}$ can be efficiently solved by taking SVD of covariance matrix of the (lifted) data points. b) The Rademacher complexity of the hypothesis class is bounded by $O(d^s/m)$ for $m$ examples. Thus by Theorem 2.1 the minimizer of ERM generalizes. The full proof is deferred to Appendix A.

## 3.2 Spectral Autoencoders

In this section we give a much broader set of hypothesis, encompassing PCA and kernel-PCA, and show how to learn them efficiently. Throughout this section we assume that the data is normalized to Euclidean norm 1, and consider the following class of hypothesis which naturally generalizes PCA:

*Definition* 3.1 (Spectral autoencoder). We define the class $\mathcal{H}_{k,s}^{\mathrm{sa}}$ as the following set of all hypothesis $(g, h)$,

$$\mathcal{H}_k^{\mathrm{sa}} = \left\{ (h, g) : \begin{array}{ll} h(x) & = Ax^{\otimes s}, A \in \mathbb{R}^{k \times d^s} \\ g(y) & = v_{\max}(By), B \in \mathbb{R}^{d^s \times k} \end{array} \right\} . \tag{3.1}$$

We note that this notion is more general than kernel PCA: suppose some $(g, h) \in \mathcal{H}_{k,s}^{\mathrm{pca}}$ has reconstruction error $\varepsilon$, namely, $A^\dagger A x^{\otimes s}$ is $\varepsilon$-close to $x^{\otimes s}$ in Euclidean norm. Then by eigenvector perturbation theorem, we have that $v_{\max}(A^\dagger A x^{\otimes s})$ also reconstructs $x$ with $O(\varepsilon)$ error, and therefore there exists a PSCA hypothesis with $O(\varepsilon)$ error as well . Vice versa, it's quite possible that for every $A$, the reconstruction $A^\dagger A x^{\otimes s}$ is far away from $x^{\otimes s}$ so that kernel PCA doesn't apply, but with spectral decoding we can still reconstruct $x$ from $v_{\max}(A^\dagger A x^{\otimes s})$ since the top eigenvector of $A^\dagger A x^{\otimes s}$ is close $x$.

Here the key matter that distinguishes us from kernel PCA is in what metric $x$ needs to be close to the manifold so that it can be reconstructed. Using PCA, the requirement is that $x$ is in Euclidean distance close to $\mathcal{M}$ (which is a subspace), and using kernel PCA $x^{\otimes 2}$ needs to be in Euclidean distance close to the null space of $c_i$'s. However, Euclidean distances in the original space and lifted space typically are meaningless for high-dimensional data since any two data points are far away with each other in Euclidean distance. The advantage of using spectral autoencoders is that in the lifted space the geometry is measured by spectral norm distance that is much smaller than Euclidean distance (with a potential gap of $d^{1/2}$). The key here is that though the dimension of lifted space is $d^2$, the objects of our interests is the set of rank-1 tensors of the form $x^{\otimes 2}$. Therefore, spectral norm distance is a much more effective measure of closeness since it exploits the underlying structure of the lifted data points.

We note that spectral autoencoders relate to vanishing component analysis [18]. When the data is close to an algebraic manifold, spectral autoencoders aim to find the (small number of) essential non-vanishing components in a noise robust manner.

## 3.3 Learnability of polynomial spectral decoding

For simplicity we focus on the case when $s = 2$. Ideally we would like to learn the best encoding-decoding scheme for any data distribution $\mathcal{D}$. Though there are technical difficulties to achieve such a general result. A natural attempt would be to optimize the loss function $f(A, B) = \|g(h(x)) - x\|^2 = \|x - v_{\max}(BAx^{\otimes 2})\|^2$. Not surprisingly, function $f$ is not a convex function with respect to $A, B$, and in fact it could be even non-continuous (if not ill-defined)!

Here we make a further realizability assumption that the data distribution $\mathcal{D}$ admits a reasonable encoding and decoding pair with reasonable reconstruction error.

*Definition* 3.2. We say a data distribution $\mathcal{D}$ is $(k, \varepsilon)$-*regularly* spectral decodable if there exist $A \in \mathbb{R}^{k \times d^2}$ and $B \in \mathbb{R}^{d^2 \times k}$ with $\|BA\|_{\mathrm{op}} \leqslant \tau$ such that for $x \sim \mathcal{D}$, with probability 1, the encoding $y = Ax^{\otimes 2}$ satisfies that

$$\mathcal{M}(By) = \mathcal{M}(BAx^{\otimes 2}) = xx^{\top} + E, \tag{3.2}$$

where $\|E\|_{\mathrm{op}} \leqslant \varepsilon$. Here $\tau \geqslant 1$ is treated as a fixed constant globally.

To interpret the definition, we observe that if data distribution $\mathcal{D}$ is $(k, \varepsilon)$-regularly spectrally decodable, then by equation (3.2) and Wedin's theorem (see e.g. [30] ) on the robustness of eigenvector to perturbation, $\mathcal{M}(By)$ has top eigenvector[2] that is $O(\varepsilon)$-close to $x$ itself. Therefore, definition 3.2 is a sufficient condition for the spectral decoding algorithm $v_{\max}(By)$ to return $x$ approximately, though it might be not necessary. Moreover, this condition partially addresses the non-continuity issue of using objective $f(A, B) = \|x - v_{\max}(BAx^{\otimes 2})\|^2$, while $f(A, B)$ remains (highly) non-convex. We resolve this issue by using a convex surrogate.

Our main result concerning the learnability of the aforementioned hypothesis class is:
*Theorem* 3.2. The hypothesis class $\mathcal{H}^{\mathrm{sa}}_{k,2}$ is $C$ - learnable with encoding length $O(\tau^4 k^4 / \delta^4)$ and bias $\delta$ with respect to $(k, \varepsilon)$-regular distributions in polynomial time.

Our approach towards finding an encoding and decoding matrice $A, B$ is to optimize the objective,

$$\text{minimize } f(R) = \mathbb{E}\left[\left\|Rx^{\otimes 2} - x^{\otimes 2}\right\|_{\mathrm{op}}\right] \tag{3.3}$$

$$\text{s.t. } \|R\|_{S_1} \leqslant \tau k$$

where $\|\cdot\|_{S_1}$ denotes the Schatten 1-norm. Suppose $\mathcal{D}$ is $(k, \varepsilon)$-regularly decodable, and suppose $h_A$ and $g_B$ are the corresponding encoding and decoding function. Then we see that $R = AB$ will satisfies that $R$ has rank at most $k$ and $f(R) \leqslant \varepsilon$. On the other hand, suppose one obtains some $R$ of rank $k'$ such that $f(R) \leqslant \delta$, then we can produce $h_A$ and $g_B$ with $O(\delta)$ reconstruction simply by choosing $A \in \mathbb{R}^{k' \times d^2} B$ and $B \in \mathbb{R}^{d^2 \times k'}$ such that $R = AB$.

We use (non-smooth) Frank-Wolfe to solve objective (3.3), which in particular returns a low-rank solution. We defer the proof of Theorem 3.2 to the Appendix A.1. With a slightly stronger assumptions on the data distribution $\mathcal{D}$, we can reduce the length of the code to $O(k^2 / \varepsilon^2)$ from $O(k^4 / \varepsilon^4)$. See details in Appendix B.

# 4 A family of optimization encodings and efficient dictionary learning

In this section we give efficient algorithms for learning a family of unsupervised learning algorithms commonly known as "dictionary learning". In contrast to previous approaches, we do not construct an actual "dictionary", but rather improperly learn a comparable encoding via convex relaxations.

We consider a different family of codes which is motivated by matrix-based unsupervised learning models such as topic-models, dictionary learning and PCA. This family is described by a matrix $A \in \mathbb{R}^{d \times r}$ which has low complexity according to a certain norm $\|\cdot\|_{\alpha}$, that is, $\|A\|_{\alpha} \leqslant c_{\alpha}$. We can parametrize a family of hypothesis $\mathcal{H}$ according to these matrices, and define an encoding-decoding pair according to

$$h_A(x) = \arg\min_{\|y\|_{\beta} \leqslant k} \frac{1}{d} |x - Ay|_1 \ , \ g_A(y) = Ay$$

We choose $\ell_1$ norm to measure the error mostly for convenience, though it can be quite flexible. The different norms $\|\cdot\|_{\alpha}, \|\cdot\|_{\beta}$ over $A$ and $y$ give rise to different learning models that have been considered before. For example, if these are Euclidean norms, then we get PCA. If $\|\cdot\|_{\alpha}$ is the max column $\ell_2$ or $\ell_{\infty}$ norm and $\|\cdot\|_b$ is the $\ell_0$ norm, then this corresponds to dictionary learning (more details in the next section).

The optimal hypothesis in terms of reconstruction error is given by

$$A^{\star} = \arg\min_{\|A\|_{\alpha} \leqslant c_{\alpha}} \mathbb{E}_{x \sim \mathcal{D}}\left[\frac{1}{d}|x - g_A(h_A(x))|_1\right] = \arg\min_{\|A\|_{\alpha} \leqslant c_{\alpha}} \mathbb{E}_{x \sim \mathcal{D}}\left[\min_{y \in \mathbb{R}^r : \|y\|_{\beta} \leqslant k} \frac{1}{d}|x - Ay|_1\right].$$

The loss function can be generalized to other norms, e.g., squared $\ell_2$ loss, without any essential change in the analysis. Notice that this optimization objective derived from reconstruction error is identically the one used in the literature of dictionary learning. This can be seen as another justification for the definition of unsupervised learning as minimizing reconstruction error subject to compression constraints.

The optimization problem above is notoriously hard computationally, and significant algorithmic and heuristic literature attempted to give efficient algorithms under various distributional assumptions(see [6, 4, 2] and the references therein). Our approach below circumvents this computational hardness by convex relaxations that result in learning a different creature, albeit with comparable compression and reconstruction objective.

### 4.1 Improper dictionary learning: overview

We assume the max column $\ell_\infty$ norm of $A$ is at most 1 and the $\ell_1$ norm of $y$ is assumed to be at most $k$. This is a more general setting than the random dictionaries (up to a re-scaling) that previous works [6, 4] studied. [3] In this case, the magnitude of each entry of $x$ is on the order of $\sqrt{k}$ if $y$ has $k$ random $\pm 1$ entries. We think of our target error per entry as much smaller than $1$[4]. We consider $\mathcal{H}_{\text{dict}}^k$ that are parametrized by the dictionary matrix $A = \mathbb{R}^{d \times r}$,

$$\mathcal{H}_k^{\text{dict}} = \left\{ (h_A, g_A) : A \in \mathbb{R}^{d \times r}, \|A\|_{\ell_1 \to \ell_\infty} \leqslant 1 \right\},$$
$$\text{where } h_A(x) = \arg\min_{\|y\|_1 \leqslant k} |x - Ay|_1 \ , \ g_A(y) = Ay$$

Here we allow $r$ to be larger than $d$, the case that is often called over-complete dictionary. The choice of the loss can be replaced by $\ell_2$ loss (or other Lipschitz loss) without any additional efforts, though for simplicity we stick to $\ell_1$ loss. Define $A^\star$ to be the the best dictionary under the model and $\varepsilon^\star$ to be the optimal error,

$$A^\star = \arg\min_{\|A\|_{\ell_1 \to \ell_\infty} \leqslant 1} \mathbb{E}_{x \sim \mathcal{D}} \left[ \min_{y \in \mathbb{R}^r : \|y\|_1 \leqslant k} |x - Ay|_1 \right] \tag{4.1}$$
$$\varepsilon^\star = \mathbb{E}_{x \sim \mathcal{D}} \left[ \tfrac{1}{d} \cdot |x - g_{A^\star}(h_{A^\star}(x))|_1 \right] \ .$$

---

**Algorithm 1** group encoding/decoding for improper dictionary learning

---

**Inputs:** $N$ data points $X \in \mathbb{R}^{d \times N} \sim \mathcal{D}^N$. Convex set $Q$. Sampling probability $\rho$.

1. **Group encoding:** Compute

$$Z = \arg\min_{C \in Q} |X - C|_1 \ , \tag{4.2}$$

and let $Y = h(X) = P_\Omega(Z)$, where $P_\Omega(B)$ is a random sampling of $B$ where each entry is picked with probability $\rho$.

2. **Group decoding:** Compute $g(Y) = \arg\min_{C \in Q} |P_\Omega(C) - Y|_1$ .

---

*Theorem* 4.1. For any $\delta > 0, p \geqslant 1$, the hypothesis class $\mathcal{H}_k^{\text{dict}}$ is $C$-learnable with encoding length $\tilde{O}(k^2 r^{1/p}/\delta^2)$, bias $\delta + O(\varepsilon^\star)$ and sample complexity $d^{O(p)}$ in time $n^{O(p^2)}$

We note that here $r$ can be potentially much larger than $d$ since by choosing a large constant $p$ the overhead caused by $r$ can be negligible. Since the average size of the entries is $\sqrt{k}$, therefore we can get the bias $\delta$ smaller than average size of the entries with code length roughly $\approx k$.

The proof of Theorem 4.1 is deferred to supplementary. To demonstrate the key intuition and technique behind it, in the rest of the section we consider a simpler algorithm that achieves a *weaker* goal: Algorithm 1 encodes *multiple* examples into some codes with the matching average encoding length $\tilde{O}(k^2 r^{1/p}/\delta^2)$, and these examples can be decoded from the codes together with reconstruction error $\varepsilon^\star + \delta$. Next, we outline the analysis of Algorithm 1, and we will show later that one can reduce the problem of encoding a single examples to the problem of encoding multiple examples.

Here we overload the notation $g_{A^\star}(h_{A^\star}(\cdot))$ so that $g_{A^\star}(h_{A^\star}(X))$ denotes the collection of all the $g_{A^\star}(h_{A^\star}(x_j))$ where $x_j$ is the $j$-th column of $X$. Algorithm 1 assumes that there exists a convex set $Q \subset \mathbb{R}^{d \times N}$ such that

$$\left\{ g_{A^\star}(h_{A^\star}(X)) : X \in \mathbb{R}^{d \times N} \right\} \subset \{ AY : \|A\|_{\ell_1 \to \ell_\infty} \leqslant 1, \|Y\|_{\ell_1 \to \ell_1} \leqslant k \} \subset Q. \tag{4.3}$$

That is, $Q$ is a convex relaxation of the group of reconstructions allowed in the class $\mathcal{H}^{\mathrm{dict}}$. Algorithm 1 first uses convex programming to denoise the data $X$ into a clean version $Z$, which belongs to the set $Q$. If the set $Q$ has low complexity, then simple random sampling of $Z \in Q$ serves as a good encoding.

The following Lemma shows that if $Q$ has low complexity in terms of sampling Rademacher width, then Algorithm 1 will give a good group encoding and decoding scheme.

*Lemma* 4.2. Suppose convex $Q \subset \mathbb{R}^{d \times N}$ satisfies condition (4.3). Then, Algorithm 1 gives a group encoding and decoding pair such that with probability $1 - \delta$, the average reconstruction error is bounded by $\varepsilon^\star + O(\sqrt{\mathcal{SRW}_m(Q)} + O(\sqrt{\log(1/\delta)/m})$ where $m = \rho N d$ and $\mathcal{SRW}_m(\cdot)$ is the sampling Rademacher width (defined in appendix), and the average encoding length is $\tilde{O}(\rho d)$.

Towards analyzing the algorithm, we will show that the difference between $Z$ and $X$ is comparable to $\varepsilon^\star$, which is a direct consequence of the optimization over a large set $Q$ that contains optimal reconstruction. Then we prove that the sampling procedure doesn't lose too much information given a denoised version of the data is already observed, and thus one can reconstruct $Z$ from $Y$.

The novelty here is to use these two steps together to denoise and achieve a short encoding. The typical bottleneck of applying convex relaxation on matrix factorization based problem (or any other problem) is the difficulty of rounding. Here instead of pursuing a rounding algorithm that output the factor $A$ and $Y$, we look for a convex relaxation that preserves the intrinsic complexity of the set which enables the trivial sampling encoding. It turns out that controlling the width/complexity of the convex relaxation boils down to proving concentration inequalities with sum-of-squares (SoS) proofs, which is conceptually easier than rounding.

Therefore, the remaining challenge is to design convex set $Q$ that simultaneously has the following properties (a) is a convex relaxation in the sense of satisfying condition (4.3). (b) admits an efficient optimization algorithm. (c) has low complexity (that is, sampling rademacher width $\tilde{O}(N \operatorname{poly}(k))$). Concretely, we have the following theorem. We note that these three properties (with Lemma 4.2) imply that Algorithm 1 with $Q = Q_p^{\mathrm{sos}}$ and $\rho = O(k^2 r^{2/p} d^{-1}/\delta^2 \cdot \log d)$ gives a group encoding-decoding pair with average encoding length $O(k^2 r^{2/p}/\delta^2 \cdot \log d)$ and bias $\delta$.

*Theorem* 4.3. For every $p \geqslant 4$, let $N = d^{c_0 p}$ with a sufficiently large absolute constant $c_0$. Then, there exists a convex set $Q_p^{\mathrm{sos}} \subset \mathbb{R}^{d \times N}$ such that (a) it satisfies condition 4.3; (b) The optimization (4.2) and (2) are solvable by semidefinite programming with run-time $n^{O(p^2)}$; (c) the sampling Rademacher width of $Q_p^{\mathrm{sos}}$ is bounded by $\sqrt{\mathcal{SRW}_m(Q)} \leqslant \tilde{O}(k^2 r^{2/p} N/m)$.

## 5 Conclusions

We have defined a new framework for unsupervised learning which replaces generative assumptions by notions of reconstruction error and encoding length. This framework is comparative, and allows learning of particular hypothesis classes with respect to an unknown distribution by other hypothesis classes. We demonstrate its usefulness by giving new polynomial time algorithms for two unsupervised hypothesis classes. First, we give new polynomial time algorithms for dictionary models in significantly broader range of parameters and assumptions. Another domain is the class of spectral encodings, for which we consider a new class of models that is shown to strictly encompass PCA and kernel-PCA. This new class is capable, in contrast to previous spectral models, learn algebraic manifolds. We give efficient learning algorithms for this class based on convex relaxations.

## Acknowledgements

We thank Sanjeev Arora for many illuminating discussions and crucial observations in earlier phases of this work, amongst them that a representation which preserves information for all classifiers requires lossless compression.

## Footnotes

[1]Technically, this is the Rademacher complexity of the class of functions $\ell \circ \mathcal{H}$. However, since $\ell$ is usually fixed for certain problem, we emphasize in the definition more the dependency on $\mathcal{H}$.

[2]Or right singular vector when $\mathcal{M}(By)$ is not symmetric

[3]The assumption can be relaxed to that $A$ has $\ell_\infty$ norm at most $k$ and $\ell_2$-norm at most $\sqrt{d}$ straightforwardly.

[4]We are conservative in the scaling of the error here. Error much smaller than $\sqrt{k}$ is already meaningful.

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
