[Supplementary Material · unsupervised_full.pdf]

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

# 6 Proof of Theorem 2.1

*Proof of Theorem 2.1.* [19, Theorem 3.1] asserts that with probability at least $1 - \delta$, we have that for every hypothesis $f \in \mathcal{H}$,

$$\underset{\mathcal{D}}{\text{loss}}(f) \leqslant \underset{S}{\text{loss}}(f) + 2\mathcal{R}_m(\mathcal{H}) + \sqrt{\frac{\log \frac{1}{\delta}}{2m}}$$

by negating the loss function this gives

$$|\underset{\mathcal{D}}{\text{loss}}(f) - \underset{S}{\text{loss}}(f)| \leqslant 2\mathcal{R}_m(\mathcal{H}) + \sqrt{\frac{\log \frac{2}{\delta}}{2m}}$$

and therefore, letting $f^* = \arg\min_{f \in \mathcal{H}} \text{loss}_{\mathcal{D}}(f)$, we have

$$\underset{\mathcal{D}}{\text{loss}}(\hat{f}_{ERM}) \leqslant \underset{S}{\text{loss}}(\hat{f}_{ERM}) + 2\mathcal{R}_m(\mathcal{H}) + \sqrt{\frac{\log \frac{1}{\delta}}{2m}} \qquad \text{(by [19, Theorem 3.1])}$$

$$\leqslant \underset{S}{\text{loss}}(f^*) + 2\mathcal{R}_m(\mathcal{H}) + \sqrt{\frac{\log \frac{1}{\delta}}{2m}} \qquad \text{( by definition of ERM)}$$

$$\leqslant \underset{\mathcal{D}}{\text{loss}}(f^*) + 6\mathcal{R}_m(\mathcal{H}) + \sqrt{\frac{4\log \frac{1}{\delta}}{2m}} \qquad \text{( using [19, Theorem 3.1] again)}$$

$\square$

## 6.1 Low reconstruction error is sufficient for supervised learning

This section observes that low reconstruction error is a sufficient condition for unsupervised learning to allow supervised learning over any future task.

*Lemma* 6.1. Consider any supervised learning problem with respect to distribution $\mathcal{D}$ over $\mathcal{X} \times \mathcal{L}$ that is agnostically PAC-learnable with respect to $L$-Lipschitz loss function $\ell$ and with bias $\gamma_1$.

Suppose that unsupervised hypothesis class $\mathcal{H}$ is $C$-learnable with bias $\gamma_2$ over distribution $\mathcal{D}$ and domain $\mathcal{X}$, by hypothesis $f : \mathcal{X} \mapsto \mathcal{Y}$. Then the distribution $\tilde{\mathcal{D}}_f$ over $\mathcal{Y} \times \mathcal{L}$, which gives the pair $(f(x), y)$ the same measure as $\mathcal{D}$ gives to $(x, y)$, is agnostically PAC-learnable with bias $\gamma_1 + L\gamma_2$.

*Proof.* Let $h : \mathcal{X} \mapsto \mathcal{Y}$ be a hypothesis that PAC-learns distribution $\mathcal{D}$. Consider the hypothesis

$$\tilde{h} : \mathcal{Y} \mapsto \mathcal{L} \ , \ \tilde{h}(y) = (h \circ g)(y)$$

Then by definition of reconstruction error and the Lipschitz property of $\ell$ we have

$$\begin{aligned}
\underset{\tilde{\mathcal{D}}_f}{\text{loss}}(\tilde{h}) &= \underset{(y,l)\sim\tilde{\mathcal{D}}_f}{\mathbb{E}} [\ell(\tilde{h}(y), l)] \\
&= \underset{(y,l)\sim\tilde{\mathcal{D}}_f}{\mathbb{E}} [\ell((h \circ g)(y), l)] \\
&= \underset{(x,l)\sim\mathcal{D}}{\mathbb{E}} [\ell(h(\tilde{x}), l)] & (\mathcal{D}(x) = \tilde{\mathcal{D}}_f(y)) \\
&= \underset{(x,l)\sim\mathcal{D}}{\mathbb{E}} [\ell(h(x), l)] + \underset{(x,l)\sim\mathcal{D}}{\mathbb{E}} [\ell(h(\tilde{x}), l) - \ell(h(x), l)] \\
&= \gamma_1 + \underset{(x,l)\sim\mathcal{D}}{\mathbb{E}} [\ell(h(\tilde{x}), l) - \ell(h(x), l)] & \text{( PAC learnability)} \\
&\leqslant \gamma_1 + L \underset{x\sim\mathcal{D}}{\mathbb{E}} \|x - \tilde{x}\| & \text{( Lipschitzness of } \ell \circ h) \\
&= \gamma_1 + L \underset{x\sim\mathcal{D}}{\mathbb{E}} \|x - g \circ f(x)\| \\
&\leqslant \gamma_1 + L\gamma_2 & \text{( } C \text{-learnability)}
\end{aligned}$$

$\square$

# A Proof of Theorem 3.1

*Proof of Theorem 3.1.* We assume without loss of generality $s = 1$. For $s > 1$ the proof will be identical since one can assume $x^{\otimes s}$ is the data points (and the dimension is raised to $d^s$).

Let $x_1, \ldots, x_m$ be a set of examples $\sim \mathcal{D}^m$. It can be shown that any minimizer of ERM

$$A^* = \underset{A \in \mathbb{R}^{d \times k}}{\arg\min} \|x_i - A^\dagger A x_i\|^2 \tag{A.1}$$

satisfies that $(A^*)^\dagger A^*$ is the the projection operator to the subspace of top $k$ eigenvector of $\sum_{i=1}^m x_i x_i^\top$. Therefore ERM (A.1) is efficiently solvable.

According to Theorem 2.1, the ERM hypothesis generalizes with rates governed by the Rademacher complexity of the hypothesis class. Thus, it remains to compute the Rademacher complexity of the hypothesis class for PCA. We assume for simplicity that all vectors in the domain have Euclidean norm bounded by one.

$$
\begin{aligned}
\mathcal{R}_S(\mathcal{H}_k^{\text{pca}}) &= \underset{\sigma \sim \{\pm 1\}^m}{\mathbb{E}} \left[ \sup_{(h,g) \in \mathcal{H}_k^{\text{pca}}} \frac{1}{m} \sum_{i \in S} \sigma_i \ell((h,g), x_i) \right] \\
&= \underset{\sigma \sim \{\pm 1\}^m}{\mathbb{E}} \left[ \sup_{A \in \mathbb{R}^{d \times k}} \frac{1}{m} \sum_{i \in S} \sigma_i \|x_i - A^\dagger A x_i\|^2 \right] \\
&= \underset{\sigma \sim \{\pm 1\}^m}{\mathbb{E}} \left[ \sup_{A \in \mathbb{R}^{d \times k}} \frac{1}{m} \sum_{i \in S} \sigma_i \mathbf{Tr}((I - A^\dagger A) \left( \sum_{i=1}^m x_i x_i^\top \right) (I - A^\dagger A)^\top) \right] \\
&= \underset{\sigma \sim \{\pm 1\}^m}{\mathbb{E}} \left[ \sup_{A \in R^{d \times k}} \mathbf{Tr} \left( (I - A^\dagger A) \left( \frac{1}{m} \sum_{i=1}^m \sigma_i x_i x_i^\top \right) \right) \right].
\end{aligned}
$$

Then we apply Holder inequality, and effectively disentangle the part about $\sigma$ and $A$:

$$
\begin{aligned}
&\underset{\sigma \sim \{\pm 1\}^m}{\mathbb{E}} \left[ \sup_{A \in \mathbb{R}^{d \times k}} \mathbf{Tr} \left( (I - A^\dagger A) \left( \frac{1}{m} \sum_{i=1}^m \sigma_i x_i x_i^\top \right) \right) \right] \\
&\leqslant \underset{\sigma \sim \{\pm 1\}^m}{\mathbb{E}} \left[ \sup_{A \in \mathbb{R}^{d \times k}} \|I - A^\dagger A\|_F \left\| \frac{1}{m} \sum_{i=1}^m \sigma_i x_i x_i^\top \right\|_F \right] && \text{(Holder inequality)} \\
&\leqslant \sqrt{d} \underset{\sigma \sim \{\pm 1\}^m}{\mathbb{E}} \left[ \left\| \frac{1}{m} \sum_{i=1}^m \sigma_i x_i x_i^\top \right\|_F \right] && (\text{ since } A^\dagger A \text{ is a projection, } \|I - A^\dagger A\| \leqslant 1.) \\
&\leqslant \sqrt{d} \underset{\sigma \sim \{\pm 1\}^m}{\mathbb{E}} \left[ \left\| \frac{1}{m} \sum_{i=1}^m \sigma_i x_i x_i^\top \right\|_F^2 \right]^{1/2} && \text{(Cauchy-Schwarz inequality)} \\
&\leqslant \sqrt{d} \sqrt{\frac{1}{m^2} \sum_{i=1}^m \langle \sigma_i x_i x_i^\top, \sigma_i x_i x_i^\top \rangle} && (\text{since } \mathbb{E}[\sigma_i \sigma_j] = 0 \text{ for } i \neq j) \\
&\leqslant \sqrt{d/m}. && (\text{by } \|x\| \leqslant 1)
\end{aligned}
$$

Thus, from Theorem 2.1 we can conclude that the class $\mathcal{H}_k^{\text{pca}}$ is learnable with sample complexity $\tilde{O}(\frac{d}{\varepsilon^2})^5$. $\qquad\square$

## A.1 Proof of Theorem 3.2

Theorem 3.2 follows from the following lemmas and the generalization theorem 2.1 straightforwardly.

*Lemma* A.1. Suppose distribution $\mathcal{D}$ is $(k, \varepsilon)$-regularly spectral decodable. Then for any $\delta > 0$, solving convex optimization (3.3) with non-smooth Frank-Wolfe algorithm [14, Theorem 4.4] with $k' = O(\tau^4 k^4 / \delta^4)$ steps gives a solution $\hat{R}$ of rank $k'$ such that $f(\hat{R}) \leqslant \delta + \varepsilon$.

*Lemma* A.2. The Rademacher complexity of the class of function $\Phi = \left\{ \left\| Rx^{\otimes 2} - x^{\otimes 2} \right\|_{\mathrm{op}} : R \text{ s.t. } \|R\|_{S_1} \leqslant \tau k \right\}$ with $m$ examples is bounded from above by at most $\mathcal{R}_m(\Phi) \leqslant 2\tau k \cdot \sqrt{1/m}$

Here Lemma A.2 follows from the fact that $\left\| Rx^{\otimes 2} - x^{\otimes 2} \right\|_{\mathrm{op}}$ is bounded above by $2\tau k$ when $\|x\| \leqslant 1$ and $\|R\|_{S_1} \leqslant \tau k$. The rest of the section focuses on the proof of Lemma A.1.

Lemma A.1 basically follows from the fact that $f$ is Lipschitz and guarantees of the Frank-Wolfe algorithm.

*Proposition* A.3. The objective function $f(R)$ is convex and 1-Lipschitz. Concretely, Let $\ell_x(R) = \|Rx^{\otimes 2} - x^{\otimes 2}\|_{\mathrm{op}}$. Then

$$\partial \ell_x \ni (u \otimes v)(x^{\otimes 2})^\top$$

where $\partial \ell_x$ is the set of sub-gradients of $\ell_x$ with respect to $R$, and $u, v \in \mathbb{R}^d$ are (one pair of) top left and right singular vectors of $\mathcal{M}(Rx^{\otimes 2} - x^{\otimes 2})$.

*Proof.* This simply follows from calculating gradient with chain rule. Here we use the fact that $A \in (\mathbb{R}^d)^{\otimes 2}$, the sub-gradient of $\|A\|_{\mathrm{op}}$ contains the vector $a \otimes b$ where $a, b$ are the top left and right singular vectors of $\mathcal{M}(A)$. We can also verify by definition that $(u \otimes v)(x^{\otimes 2})^\top$ is a sub-gradient.

$$
\begin{aligned}
f(R') - f(R) &\geqslant (u \otimes v)^\top (R' x^{\otimes 2} - R x^{\otimes 2}) && \text{(by convexity of } \|\cdot\|_{\mathrm{op}}\text{)} \\
&= \langle (u \otimes v)(x^{\otimes 2})^\top, R' - R \rangle .
\end{aligned}
$$

$\square$

Now we are ready to prove Lemma A.1.

*Proof of Lemma A.1.* Since $\mathcal{D}$ is $(k, \varepsilon)$-regularly decodable, we know that there exists a rank-$k$ solution $R^*$ with $f(R^*) \leqslant \varepsilon$. Since $\|R^*\|_{\mathrm{op}} \leqslant \tau$, we have that $\|R\|_{S_1} \leqslant \mathrm{rank}(R^*) \cdot \|R\|_{\mathrm{op}} \leqslant \tau k$. Therefore $R^*$ is feasible solution for the objective (3.3) with $f(R^*) \leqslant \varepsilon$.

By Proposition A.3, we have that $f(R)$ is 1-Lipschitz. Moreover, for any $R, S$ with $\|R\|_{S_1} \leqslant \tau k, \|S\|_{S_1} \leqslant \tau k$ we have that $\|R - S\|_F \leqslant \|R\|_F + \|S\|_F \leqslant \|R\|_{S_1} + \|S\|_{S_1} \leqslant 2\tau k$. Therefore the diameter of the constraint set is at most $\tau k$.

By [14, Theorem 4.4], we have that Frank-Wolfe algorithm returns solution $R$ with $f(R) - f(R^*) \leqslant \varepsilon + \delta$ in $\left( \frac{\tau k}{\delta} \right)^4$ iteration.

$\square$

# B   Shorter codes with relaxed objective for Polynomial Spectral Components Analysis

**Notations.**   For a matrix $A$, let $\sigma_1(A) \geqslant \sigma_2(A) \geqslant ..$ be its singular values. Then the Schatten p-norm, denoted by $\|\cdot\|_{S_p}$, for $p \geqslant 1$ is defined as $\|A\|_{S_p} = \left( \sum_i \sigma_i(A)^p \right)^{1/p}$. For even integer $p$, an equivalent and simple definition is that $\|A\|_{S_p}^p \triangleq \mathbf{Tr}((A^\top A)^{p/2})$.

In this section we consider the following further relaxation of objective (3.3).

$$\text{minimize } f_4(R) := \mathbb{E}\left[ \left\| Rx^{\otimes 2} - x^{\otimes 2} \right\|_{S_p}^2 \right] \tag{B.1}$$

$$\text{s.t. } \|R\|_{S_1} \leqslant \tau k$$

Since $\|A\|_F \geqslant \|A\|_{S_4} \geqslant \|A\|_{S_\infty} = \|A\|$, this is a relaxation of the objective (3.3), and it interpolates between kernal PCA and spectral decoding. Our assumption is weaker than kernal PCA but stronger than spectral decodable.

*Definition* B.1 (Extension of definition 3.2). We say a data distribution $\mathcal{D}$ is $(k, \varepsilon)$-*regularly* spectral decodable with $\|\cdot\|_{S_p}$ norm if the error $E$ in equation (**??**) is bounded by $\|E\|_{S_p} \leqslant \varepsilon$.

We can reduce the length of the code from $O(k^4)$ to $O(k^2)$ for any constant $p$.

*Theorem* B.1. Suppose data distribution is $(k, \varepsilon)$-spectral decodable with norm $\|\cdot\|_{S_p}$ for $p = O(1)$, then solving (B.1) using (usual) Frank-Wolfe algorithm gives a solution $\hat{R}$ of $k' = O(k^2\tau^2/\varepsilon^2)$ with $f(R) \leqslant \varepsilon + \delta$. As a direct consequence, we obtain encoding and decoding pair $(g_A, h_B) \in \mathcal{H}_{k'}^{\text{sa}}$ with $k' = O(k^2\tau^2/\varepsilon^2)$ and reconstruction error $\varepsilon + \delta$.

The main advantage of using relaxed objective is its smoothness. This allows us to optimize over the Schatten 1-norm constraint set much faster using usual Frank-Wolfe algorithm. Therefore the key here is to establish the smoothness of the objective function. Theorem B.1 follows from the following proposition straightforwardly.

*Proposition* B.2. Objective function $f_p$ (equation (B.1)) is convex and $O(p)$-smooth.

*Proof.* Since $\|\cdot\|_{S_p}$ is convex and composition of convex function with linear function gives convex function. Therefore, $\left\|Rx^{\otimes 2} - x^{\otimes 2}\right\|_{S_p}$ is a convex function. The square of an non-negative convex function is also convex, and therefore we proved that $f_p$ is convex. We prove the smoothness by first showing that $\|A\|_{S_p}^2$ is a smooth function with respect to $A$. We use the definition $\|A\|_{S_p}^p = \mathbf{Tr}((A^\top A)^{p/2})$. Let $E$ be a small matrix that goes to 0, we have

$$\|A + E\|_{S_p}^p = \mathbf{Tr}((A^\top A)^{p/2}) + T_1 + T_2 + o(\|E\|_F^2) \tag{B.2}$$

where $T_1, T_2$ denote the first order term and second order term respectively. Let $U = A^\top E + E^\top A$ and $V = A^\top A$. We note that $T_2$ is a sum of the traces of matrices like $UVUVU^{p/2-2}$. By Lieb-Thirring inequality, we have that all these term can be bounded by $\mathbf{Tr}(U^{p/2-2}V^2) = 2\mathbf{Tr}((A^\top A)^{p/2-2}A^\top EE^\top A) + 2\mathbf{Tr}((A^\top A)^{p/2-2}A^\top EA^\top E^\top)$. For the first term, we have that

$$\mathbf{Tr}((A^\top A)^{p/2-2}A^\top EE^\top A) \leqslant \|(AA^\top)^{(p-2)/4}E\|^2 \leqslant \|(AA^\top)^{(p-2)/4}\|_{S_\infty}^2\|E\|_F^2 = \|A\|_{S_\infty}^{p-2}\|E\|_F^2$$

where in the first inequality we use Cauchy-Schwarz. Then for the second term we have

$$\mathbf{Tr}((A^\top A)^{p/2-2}A^\top EA^\top E^\top) \leqslant \|(A^\top A)^{(p-2)/4}E\|_F\|AE(A^\top A)^{(p-4)/4}\|_F$$

$$\leqslant \|(A^\top A)^{(p-2)/4}E\|_F^2 \qquad \text{(by Lieb-Thirring inequality)}$$

$$\leqslant \|(AA^\top)^{(p-2)/4}\|^2\|E\|_F^2 = \|A\|_{S_\infty}^{p-2}\|E\|_F^2 \tag{B.3}$$

Therefore, finally we got

$$T_2 \leqslant O(p^2) \cdot \|A\|_{S_{p-2}}^{p-2}\|E\|_F^2 \tag{B.4}$$

Therefore, we complete the proof by having,

$$\|A + E\|_{S_p}^2 \leqslant (\|A\|_{S_p}^p + T_1 + T_2 + o(\|E\|^2))^{2/p} \leqslant \|A\|_{S_p}(1 + T_1' + \frac{2}{p\|A\|_{S_p}^{p/2}}T_2) + o(\|E\|^2)$$

$$\text{(by } (1+x)^{p/2} \leqslant 1 + 2x/p + o(\|x\|^2))$$

$$\leqslant \|A\|_{S_p}^2 + T_1'' + O(p)\|E\|^2 + o(\|E\|^2) \qquad \text{(by equation (B.3))}$$

$\square$

# C  Toolbox

*Lemma* C.1. Let $p \geqslant 2$ be a power of 2 and $u = [u_1, \ldots, u_n]$ and $v = [v_1, \ldots, v_n]$ be indeterminants. Then there exists SoS proof,

$$\vdash \left(\sum_j u_i v_j^{p-1}\right)^p \leqslant \left(\sum_i u_i^p\right)\left(\sum v_i^p\right)^{p-1} \tag{C.1}$$

*Proof Sketch.* The inequality follows from repeated application of Cauchy-Schwarz. For example, for $p = 4$ we have

$$\vdash \left(\sum_i u_i^4\right)\left(\sum v_i^4\right)^3 \geqslant \left(\sum_i u_i^2 v_i^2\right)^2\left(\sum v_i^4\right)^2 \qquad \text{(by Cauchy-Schwarz)}$$

$$\geqslant \left( \sum_i u_i v_i^3 \right)^4 \qquad \text{(by Cauchy-Schwarz again)}$$

For $p = 2^s$ with $s > 2$, the statement can be proved inductively. $\qquad\square$