[Reviews · NeurIPS 2016]

Reviewer 1

Summary

The paper proposes a unified theory for unsupervised learning in terms of encoding and decoding functions. This framework is instantiated on two types of problems (PCA-type) and dictionary learning. In both, cases the paper gives sample complexity guarantees for learning these models, although both results don't exactly fall into the framework (requiring more assumptions or modified definitions).

Qualitative Assessment

This paper overpromises and underdelivers in my opinion. The introduction claims that this approach to unsupervised learning removes generative assumptions that have been common in the area. I do agree that the unified formulation has many desirable properties, including the notion of excess risk and the lack of assumptions on the data generating process. However, for the PSCA problem, the paper does make a generative assumption, namely the regularly spectral decodable assumption. And for the dictionary learning problem, the paper changes the formulation somewhat substantially to allow for group encoding/decoding. So the paper fails to provide strong evidence supporting the unified view of unsupervised learning through CONUS-learnability. Both interesting examples provided in the paper are not examples of this property as they require slight variations. I am worried about Definition 3.1 for many reasons. First, I think there is serious dimension mismatch in the definition. Either we should take x^{\otimes l} or A should be in \RR^{k \times d^{2}}. For the purposes of this discussion, fix l=2 so the dimensions here at least check out. Now I'm supposed to view By \in \RR^{d^2}, which means that v_max(By) \in \RR^d, which cannot be compared to x^{\otimes 2} as is required in the loss function. So how am I supposed to think about this? Are we supposed to take v_max(By)^{\otimes 2} to get the decoding? I do see a difference between PCA and PSCA, but it seems to be that the decoding matrix need not be A^\dagger. If you set B = A^\dagger then, at least intuitively, you should recover PCA, since all of the matrices here are rank one, taking the top singular vector shouldn't do anything. Of course because of errors in dimension and the issues I've alluded to, I cannot tell exactly what is going on. The dictionary learning result seems interesting, although I am not an expert on the problem. I do know that others have used Sum-of-Squares for this problem, and I wonder how the proposed algorithm and results compare ([5,19]). I think the paper can be significantly improved by focusing exclusively on the dictionary learning problem, and including a detailed comparison to related work. This paper is riddled with minor errors that on one hand do not affect the high-level message, but on the other hand to do make the details unclear. For example, 1. The definition of CONUS-learnability has an unbound parameter. What is d? Is it n? 2. Is the sample complexity not allowed to depend on k? This seems strange to me. Many generative approaches for clustering have sample complexity that depend on the number of clusters, which I believe would be related to k. For example, to map k-means into the set up here, I would set g_A(x) = \argmin_i ||x - a_i||^2 and h_A(i) = a_i, where A \in \RR^{d x k} is a matrix whose columns are the cluster centers. In this example, the embedding dimension is at least log(k), although k might be more natural if we use one-hot encoding. 3. In section 3.1 I believe A should be in R^{k times d^l}, otherwise the dimensions don't work out in the matrix vector product defining \Hcal_{k,l}^{pca}.

Confidence in this Review

2-Confident (read it all; understood it all reasonably well)


Reviewer 2

Summary

The paper suggests a (PAC inspired) framework for studying certain class of unsupervised learning problems. In this framework, the algorithm is given examples from some distribution on R^d, and its goal is to find a pair of functions f:R^d -> R^k, g:R^k -> R^d (where k is small) such that |x-g(f(x))| is small for most examples x. The way algorithms are measured is by comparing the performance of the returned hypothesis (pair of functions) to the best hypothesis in some predefined hypothesis class. (the paper considers a more abstract and general setting then these, but for simplicity I'm concentrating on those). Several supervised learning problems naturally fit into this framework, such as PCA and dictionary learning. Like the PAC model for supervised learning, this framework constitutes a common ground for talking about same complexity, computational complexity and so on. Interestingly, improper learning makes sense in this framework. The authors demonstrate the merits of this framework by exhibiting two algorithms that generalize kernel PCA and dictionary learning (which is intractable for proper learning). These algorithms are based on spectral techniques.

Qualitative Assessment

I find the framework quite interesting. It is attractive from theoretical standpoint, as it makes no generative assumptions. In addition, it might pave the way to new unsupervised learning algorithms (as demonstrated in this paper). That being said, I have reservation with the presentation of this frame work as a new and general framework for unsupervised learning. First, without the issue of improper learning, the current framework is subsumed by Vapnik's general model of learning. As for improper learning, I find it much less natural for unsupervised learning (compared to supervised learning). The main reason is that while in supervised learning minimizing the predefined loss is usually the final goal, in unsupervised learning the goal is more fuzzy, and can be described as "learning a better representation of the data, so that simpler classifiers would be effective". Minimizing the loss function over a specific set of mappings (say, linear functions) is done because the intuition of the user is that finding such a mapping will reach that goal. Now, if this class of function is changed (with preservation of the bit complexity of the output), this might change that belief (think of applying random permutation to the output space (Y in the paper).

Confidence in this Review

2-Confident (read it all; understood it all reasonably well)


Reviewer 3

Summary

This paper first gives a definition of unsupervised learnability (in the same sense that PAC learning is a definition of supervised learnability), and shows under this definition how "dictionary learning" is learnable. This stands in constrast to the existing mathematical literature on dictionary learning, which focuses not on minimizing a loss, but instead on parameter recovery, a task which requires significantly more assumptions. The definition of unsupervised learning is based on reconstruction and compression: basically, we can do well on an unsupervised task if we have a pair of mappings, a compressor to k bits and a decompressor, whose composition gives low reconsutrction error. The dictionary learning solver uses SDP hierarchies (SoS/Parrilo/Lasserre).

Qualitative Assessment

I am torn about this paper. On the plus side: + the paper tackles the question of formally defining unsupervised learning, and packages it with a solution, in this framework, to a popular problem. Though it is easy to come up with definitions and moreover the one in the paper is not only natural but can be seem as derived/influenced from other sources, it is done fairly well and the algorithm it is packaged with is interesting. + In recent years, the theoretical community has seemed to shift to parameter estimation, however the applied community seems to keep caring about prediction error. This rift just means that more people view theory as inconsequential, therefore I think it is important to attack this problem and bring attention to it, even if the attempt is flawed (and the attempt here is not flawed, it is a rare use of SoS in machine learning). On the minus side: - The paper was obviously rushed. I don't just mean small things like typos, but rather the following structural/presentation issues: 1. The centeral definition, "CONUS", is not even satisfied by the main algorithm (algorithm 1). The fix is obvious: the definition should allow for randomization in the algorithm, not only in the data. Why wasn't the original definition worded this way? Why wasn't the new definition (in section 4) foreshadowed? Why is Definition 4.1 so sloppily written? Is it in expectation or with high probability? 2. I think space is suboptimally organized. For instance, there is no description of the actual solver in the body of the paper. This was really less important than section 5, and all the details in subsections 3.2 and 3.3? Reviewers are supposed to be able to reasonably verify the paper without looking at the appendix (i.e., there are 50 page nips papers, but the key nuggets should be in the body). It is especially strange to me that dictionary learning is in the title, yet subsections 3.2 and 3.3 have the algorithm and analysis specified in more detail. Some detailed comments: Line 29. Well, Lasso needs many assumptions. Line 31, also subsection 1.1. What about clustering? There the guarantees are cost-based and not recovery-based. E.g., kmeans. Line 56. PAC learning is distribution free. Line 85. basis -> basic. Line 89, item "3.": An decodable -> A decodable. Also, item 3 has a grammar error. Line 90, item "4.". The claim does not match appendix A.1, which is only about Lipschitz losses. Learning proceeds just fine with non-lipschitz losses, non-smooth losses, etc., even without any sort of explicit or implicit lipschitz-ing / smooth-ing technique. Paragraph above line 91. This seems the natural place to put randomization over the algorithm, and that would unify things with section 4, right? At least, section 4's alternate definition should be foreshadowed. Line 100. This should be clarified, since I can pack all turing machines into a single real number. Line 126. pretended -> prepended. Line 130. "can efficiently solvable". First line subsection 4.1. subsequence -> subsequent. I noticed this at least one other place as well. equation display above algorithm 1. double comma. Stage 1 of algorithm 1. It was not clear at first reading that this was a random sparsification. Line 208. "Holds in average" is unclear, as discussed above. Line 223. "first use" -> "first uses". Line 467. This is subjective/stylistic, but may I suggest also citing Parrilo, who often seems to receive less credit than Lasserre?

Confidence in this Review

2-Confident (read it all; understood it all reasonably well)


Reviewer 4

Summary

This paper proposes a PAC-style learning framework for unsupervised learning which the authors call CONUS (Comparative Non-generative UnSupervised) and also proposes learning methodology. In contrast to PAC-learning, the CONUS framework is distribution-dependent hence suitable to analysis of unsupervised learning. To further support such analysis, it is non-generative, i.e. makes no assumption on joint distributions but instead evaluates learning by minimizing reconstruction error. As an illustrative example the authors show how Principle Component Analysis can be formally learned in their setting. They also give a general class of hypothesis classes for unsupervised learning which they call polynomial spectral decoding, and they show how they can be efficiently learned under their definitions via convex optimization. Their main contribution is a convex optimization-based methodology for improper learning of a class of hypothesis classes, which includes dictionary learning. For analysis of this they introduce an averaged variant on their original CONUS-learnable definition. Their new class of algorithms are polynomial time for a broad class of dictionary models, thus significantly advancing state of the art.

Qualitative Assessment

* I need to clarify: my low score on question 8 is only because the paper must be revised to fix very frequent typos and English issues (i.e., in its current form the paper's presentation is not NIPS-acceptable just because of these issues). From a technical point of view however the paper is clear and very well-written and I feel it makes an important contribution. My score on question 8 would have been a 4 if the English issues were not present. I therefore strongly vote for acceptance after thorough revision. * It is good that sufficient care is taken to provide background for non-experts (see some positive comments on this later on) but in some places at the very start the level of discussion is aimed a bit low, for example in lines 14-19. * It would be essential that the authors have the paper proofread for style and grammar issues. Currently, there are too many errors. Some examples (likely careless typos): a missing "is" in the sentence "it does not assume any generative model and based on..." in the abstract; should be "learner" or "learning algorithm"in the sentence "In PAC learning, the learning can access.."; the sentence "Another approach from the information theory literature studies with online lossless compression." does not make sense as stated but I'm not sure what the authors intended; "crucial" twice in a row around line 90; should be "if there exists" in line 96; "the function that compute" should be "... computes", "can efficiently solvable" in line 130, "matrice" should be "matrix" etc... * Another (relatively minor) presentation issue: the chosen typesetting/formatting style italicizes the words Definition and Theorem etc.. but leaves the body of these environments un-italicized. This does not seem standard in NIPS; more common and clearer in my opinion is boldface for Def/Thm and italics for body (also italics for the word Proof not boldface). Was the submitted version generated with the NIPS style file? Please verify. Also, when referring to a specific definition or theorem, I believe more common NIPS practice is to capitalize the word Definition/Theorem, e.g. "from Theorem X we see..." * Line 111, there is an errant period: "Notations: . " * "We keep a list of notations that we are using." - > change to for example "The following is a listing of notation that we use." * Line 126 "pretended to the new data points" should be changed. Also, state what \ell is here, or perhaps avoid this letter altogether since it is also used for the loss function. * The actual specification of the framework and discussion around it in Section 2 is well-done (in particular, here the authors assume just enough familiarity with standard theory and give a solid high-level explanation of why Theorem 2.1 holds based on this). Also high-level explanation for the proof of Theorem 3.1 is appreciated and convincing. The framework is very promising. * The example and variant of framework in Section 4 on optimization encoding and efficient dictionary learning is nice. It is well-described. The authors also take care to explain its element of novelty on lines 231ff. The clearly presented proof overview which follows is greatly appreciated. It seems to correct to me, though I did not have time to read the detailed proof in the Appendix. * I am curious if the authors have considered auto encoders in the context of the CONUS framework. * Throughout the paper, the authors need to refer to a class of hypothesis class or a type of hypothesis class. I think it would be good to fix handling/wording for this at the start and consistently use it also in defining the framework in Section 2 + later discussions. As it is, one sees the wording "hypothesis class" in the framework definition, but "set of hypothesis" to refer to a type of hypothesis class on line 134. * The example and discussion in line 144ff is nice.

Confidence in this Review

2-Confident (read it all; understood it all reasonably well)


Reviewer 5

Summary

The Authors have claimed to find a new framework to understand unsupervised learning which both generalizes well known such problems (e.g. PCA, dictionary learning etc.), and furthermore some problems which are intractable in general can be solved efficiently under these settings. This framework considers the following setting. Let X,Y be some spaces (usually Euclidean spaces) and let H be a set of pairs of functions (h,g) where h:X->Y and g:Y->X. Given a point x in X and a pair (h,g) we have the loss function l(h,g,x)>=0. The problem is given a distribution D on X and an i.i.d sequence (x_1,...,x_n) chosen according to D, find a pair (h,g) that minimizes E(l(h,g,x)) where x distributes according to D. Furthermore, we assume that h has a small representation as a function. This definition is an adaptation of the standard PAC learning definition, and it can be shown that if the Rademacher complexity is small then the ERM hypothesis will always produce a good hypothesis (namely, comparable to the best hypothesis).

Qualitative Assessment

The paper itself suffer from two main problems. The first is the lack of well written definitions. In line 89 (3), what is the hypothesis? is it h\g\(h,g)? Another example, in Definition 2.1 (line 95) the sample size m(epsilon,delta) is a function of d which is not a parameter of the definition (maybe it should be caligraphic D?). Later on they say that "h(x) has explicit representation with length at most k bits", and one line later they write that this "explicit representation" will use real numbers instead of bits. Furthermore, there is no explanation why the representation of h should be bounded but not the representation of g. The examples of PCA and kernel PCA which should give the intuition for the example are also not clear (section 3.1). In the PCA example, the class doesn't "operates" and rather the linear functions in the hypotheses operate. In the PCA kernel example the meaning of the expression "the tensor power of the data is pretended to the new data points." is not clear. Similarly the definition of H_k,l^pca is not clear - is A fixed? what is d? what is Ax^l if A has d columns and x in R^n? In section 3.2 the class H_k,l^psca is defined (without the l parameter) and again we have the parameter d which is undefined, the space from which x,y are taken are not defined (namely what are X and Y), and the tensor power of x should probably be l and not 2. If the main goal of a paper is to give a new set of definitions in order to get a new insight to some problems, it should have a clear well written definitions and well written examples for intuition. The second problem is that the authors do not mention Vapnik's general learning setting. This setting is very close to the definition they give and should be at least mentioned if not built upon in such a paper. Other than these two issues there are several typos and unclear sentences. For example, in lines 4-5 in the abstract they write "this allows to avoid known computational results and algorithms based on convex relaxations". Do we want to allow these algorithms or avoid them? line 59 - "learned" instead of "leaned" line 82 - " a specific" instead of "e specific" line 89 - "A decodable" instead of "An deocodable" line 96 - "If there exists" instead of "if exists"

Confidence in this Review

1-Less confident (might not have understood significant parts)


Reviewer 6

Summary

This paper proposes a new framework for considering unsupervised learning algorithm. To give an example, in the case of e.g. non-negative matrix factorization where we might need to solve an NP-hard problem, we could instead attempt to perform "improper learning" by simply finding a projection of the data that allows efficient reconstruction. More generally, the goal is to find a projection of the data onto k-bit representations, such that the data can be constructed almost as well as any reconstruction scheme in a given hypothesis class H. This framework is applied in the settings of PCA and dictionary learning.

Qualitative Assessment

This was the most interesting paper I reviewed at NIPS this year, in two ways: first, the ideas in the paper were quite intriguing; secondly, I wasn't really sure what to do with it as a reviewer. I thought the paper had both a lot of strengths and a lot of weaknesses, and while the scores above are my best attempt to turn these strengths and weaknesses into numerical judgments, I think it's important to consider the strengths and weaknesses holistically when making a judgment. Below are my impressions. First, the strengths: 1. The idea to perform improper unsupervised learning is an interesting one, which allows one to circumvent certain NP hardness results in the unsupervised learning setting. 2. The results, while mostly based on "standard" techniques, are not obvious a priori, and require a fair degree of technical competency (i.e., the techniques are really only "standard" to a small group of experts). 3. The paper is locally well-written and the technical presentation flows easily: I can understand the statement of each theorem without having to wade through too much notation, and the authors do a good job of conveying the gist of the proofs. Second, the weaknesses: 1. The biggest weakness is some issues with the framework itself. In particular: 1a. It is not obvious that "k-bit representation" is the right notion for unsupervised learning. Presumably the idea is that if one can compress to a small number of bits, one will obtain good generalization performance from a small number of labeled samples. But in reality, this will also depend on the chosen model class used to fit this hypothetical supervised data: perhaps there is one representation which admits a linear model, while another requires a quadratic model or a kernel. It seems more desirable to have a linear model on 10,000 bits than a quadratic model on 1,000 bits. This is an issue that I felt was brushed under the rug in an otherwise clear paper. 1b. It also seems a bit clunky to work with bits (in fact, the paper basically immediately passes from bits to real numbers). 1c. Somewhat related to 1a, it wasn't obvious to me if the representations implicit in the main results would actually lead to good performance if the resulting features were then used in supervised learning. I generally felt that it would be better if the framework was (a) more tied to eventual supervised learning performance, and (b) a bit simpler to work with. 2. I thought that the introduction was a bit grandiose in comparing itself to PAC learning. 3. The main point (that improper unsupervised learning can overcome NP hardness barriers) didn't come through until I had read the paper in detail. When deciding what papers to accept into a conference, there are inevitably cases where one must decide between conservatively accepting only papers that are clearly solid, and taking risks to allow more original but higher-variance papers to reach a wide audience. I generally favor the latter approach, I think this paper is a case in point: it's hard for me to tell whether the ideas in this paper will ultimately lead to a fruitful line of work, or turn out to be flawed in the end. So the variance is high, but the expected value is high as well, and I generally get the sense from reading the paper that the authors know what they are doing. So I think it should be accepted. Some questions for the authors (please answer in rebuttal): -Do the representations implicit in Theorems 3.2 and Theorem 4.1 yield features that would be appropriate for subsequent supervised learning of a linear model (i.e., would linear combinations of the features yield a reasonable model family)? -How easy is it to handle e.g. manifolds defined by cubic constraints with the spectral decoding approach?

Confidence in this Review

2-Confident (read it all; understood it all reasonably well)